# A Novel Hybrid CSP-PV Power Plant Based on Brayton Supercritical CO$_2$ Thermal Machines

José Ignacio Linares [1,*], Arturo Martín-Colino [1], Eva Arenas [1,2], María José Montes [3], Alexis Cantizano [1,2] and José Rubén Pérez-Domínguez [1]

1 Rafael Mariño Chair on New Energy Technologies, Comillas Pontifical University, Alberto Aguilera 25, 28015 Madrid, Spain; 201908502@alu.comillas.edu (A.M.-C.); earenas@comillas.edu (E.A.); alexis.cantizano@comillas.edu (A.C.); jrpdominguez@comillas.edu (J.R.P.-D.)
2 Institute for Research in Technology, ICAI, Comillas Pontifical University, Santa Cruz de Marcenado 26, 28015 Madrid, Spain
3 Department of Energy Engineering, Universidad Nacional de Educación a Distancia (UNED), Juan del Rosal 12, 28040 Madrid, Spain; mjmontes@ind.uned.es
* Correspondence: linares@comillas.edu

**Abstract:** A novel hybrid CSP-PV power plant is presented. Instead of the integration used in current hybrid power plants, where part of the PV production is charged into the thermal energy storage system through electrical resistors, the proposed system integrates both PV and thermal solar fields using a high-temperature heat pump. Both the heat pump and the heat engine are based on Brayton supercritical CO$_2$ thermodynamic cycles. Such integration allows for charging the molten salt storage as if a central tower receiver field supplied the thermal energy, whereas parabolic trough collectors are employed. Unlike conventional hybrid plants, where the storage of PV production leads to a decrease in power injected into the grid throughout the day, the power injected by the proposed system remains constant. The heat engine efficiency is 44.4%, and the COP is 2.32. The LCOE for a 50 MWe plant with up to 12 h of storage capacity is USD 171/MWh, which is lower than that of existing CSP power plants with comparable performance. Although the cost is higher compared with a PV plant with batteries, this hybrid system offers two significant advantages: it eliminates the consumption of critical raw materials in batteries, and all the electricity produced comes from a synchronous machine.

**Keywords:** Brayton supercritical CO$_2$ power cycle; high-temperature heat pump; thermal energy storage; Carnot battery; CSP-PV hybrid power plant



## 1. Introduction

To stay on track to limit the global temperature rise to 1.5 °C above pre-industrial levels, the world needs a significant increase in renewable energy. By the end of 2022, renewables comprised 40% of the global installed power capacity. This represents the largest increase in renewable energy capacity to date, adding nearly 295 GW and boosting the renewable energy stock by 9.6%, with solar energy contributing almost two-thirds [1].

Integrating large-scale storage systems becomes crucial to ensure a stable and reliable energy supply as the world seeks to reduce carbon emissions and move towards cleaner energy sources. Storage systems enable the capture and efficient use of excess energy generated by intermittent renewable sources, such as photovoltaic (PV) and wind power. In concentrated solar power (CSP) systems, thermal energy storage (TES) technology has proven particularly effective in converting intermittent solar power into a manageable source. Hybridizing CSP technology with other energy sources enables even better performance by balancing supply and demand, reducing intermittent generation, and lowering the cost of electricity. Thus, integrating storage systems in renewable energy plants, such as hybrid CSP plants, is vital in creating a sustainable and resilient energy infrastructure,

supporting decarbonization goals and providing a cost-effective, reliable, and manageable energy source.

CSP has traditionally been considered more suitable than PV for baseload generation because thermal storage is more economical than battery storage. However, the solar fields used in CSP technology are usually relatively expensive. In addition, PV plants without storage supply electricity at a much lower cost than CSP plants of comparable capacity without storage. Therefore, the integration of these two technologies is presented as an attractive approach to achieve baseload solar generation with an affordable levelized cost of electricity (LCOE).

Figure 1 shows a standard hybridization scheme between a PV and a CSP system [2]. The key idea is to maximize power generation and reduce the LCOE by ensuring coverage over a significant portion of the day. To accomplish this goal, the PV plant, characterized by its low LCOE [3], is strategically designed to generate power during hours of solar radiation. On the other hand, the CSP system captures and stores the thermal energy produced by the solar field. This stored thermal energy is then converted into electricity during hours when solar radiation is unavailable. This innovative approach eliminates the need for expensive battery storage [4] and avoids the necessity of oversizing the solar field. This is because it only needs to generate enough thermal energy to be converted into electricity during periods without solar radiation. Heating resistors might be placed into the thermal energy storage system (typically a molten-salt two-tank configuration), powered by the PV plant, ensuring that excess energy is utilized. However, it is important to emphasize that the efficiency of the heat engine significantly impacts the power recovery from the stored PV energy. In the best-case scenario, approximately half of the stored power is typically recovered.

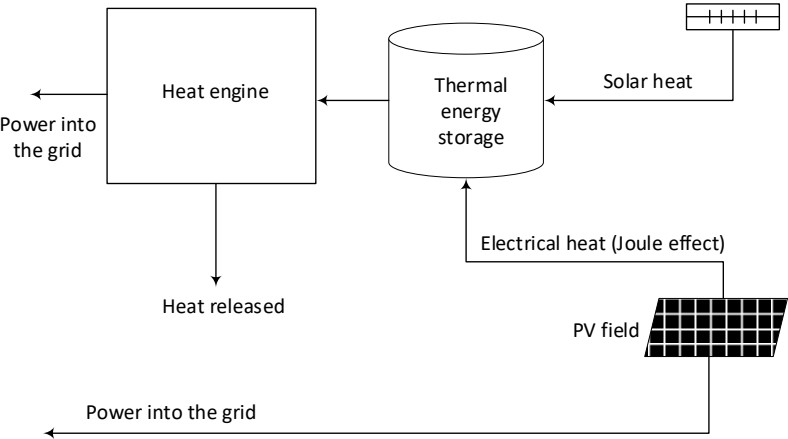

**Figure 1.** Conventional scheme of hybrid CSP-PV plants.

The optimal design of highly integrated CSP-PV power plants was addressed in [5]. A comparison was made between grid-level integration and the inclusion of electric heaters to convert excess PV electricity into heat by storing it in the CSP. The results concluded that the hybrid CSP-PV technology achieved a lower cost of electricity for dispatch levels above 50% and that designs with electric heaters had a 3.6% to 10% lower LCOE compared with grid integration. In [6], it was shown that the hybridization of CSP plants with PV systems could significantly increase the overall plant capacity factor, which contributes to achieving a fully dispatchable solar electricity production system. The study found that CSP-PV plants can achieve capacity factors of 80% or more while reducing the size of the CSP solar field. This was achieved while maintaining a high capacity factor and reducing the LCOE in the range of 4% to 7% for plants with parabolic trough collector (PTC) technology and between 1.5% and 4% for plants with central receiver system (CRS) technology. In addition, a reduction in solar field size of approximately 40% was observed for PTC hybrid plants and 30% for CRS hybrid plants. In a similar analysis, in South Africa [7], capacity factors of up

to 90% were obtained, with an LCOE of USD 133–157/MWh. In [8], typical capacity factors for intermittent renewable energy sources were revealed to be in the 20% to 40% range. However, the Solar Reserve's Crescent Dunes project achieved a capacity factor of over 80%, and when combined with PV systems, the capacity factor rose to approximately 90%. In [9], a methodology for designing and sizing hybrid CSP-PV plants was described, which used a transient simulation model coupled with an evolutionary optimization algorithm. The results showed that the capacity factor reached values above 85% and the LCOE was lower than stand-alone CSP plants. In [10], a case study that calculated the LCOE for a hybrid CSP-PV plant at the Atacama Solar Platform was presented. The objective was to evaluate new options for continuous power delivery. The results indicated that CSP-PV plants are a feasible option that can contribute to the continuous delivery of sustainable electricity. Two LCOE scenarios, based on IEA studies, were evaluated between 2014 and 2050, obtaining values of USD 146.9 and 85.7/MWh and USD 138.8 and 77.4/MWh, respectively. The integration of CSP-PV plants in two Saudi Arabian cities, Riyadh and Tabuk, was analyzed in [11]. By setting a capacity factor of 79%, it was found that a solar multiple of 6 in Riyadh and 3.5 in Tabuk was required for a single solar plant. However, with the introduction of the hybrid concept, the solar multiple was significantly reduced. It ranged from 2.9 to 3 in Riyadh and from 1.78 to 1.85 in Tabuk.

A review of advanced power cycles to improve efficiency and reduce costs was carried out in [12]. It was mentioned that subcritical steam turbines are a developed option, but more agility and flexibility in their operation are required. On the contrary, supercritical steam turbines are considered interesting but are generally too large for existing solar towers. Closed Brayton cycle systems with supercritical $CO_2$ (S-CO2) are in the early stages of development but offer the promise of high efficiency at reasonable temperatures and in various capacities, with the prospect of significantly reduced costs. In [13], different supercritical $CO_2$ power cycle configurations in a hybrid CSP-PV plant with salt storage were analyzed. The scalability of the plant was investigated, obtaining an LCOE below EUR 66/MWh and capacity factors above 70% for a capacity of 100 MWe. In locations with high solar irradiation, a capacity factor of 85% and an LCOE of EUR 46/MWh was achieved. In addition, no significant differences in terms of the S-CO2 power cycle configuration were observed. A hybrid CSP-PV-wind plant based on the S-CO2 Brayton cycle was proposed for different load demand scenarios [14]. It was proven that the load demand scenarios significantly impact load matching and economic performance. The system can meet more than 90% of the annual load demand. The LCOE reached USD 216.9/MWh for the load following scenario, being 32.1% higher than in stable production scenarios.

Hybridization of CSP and PV systems can include thermal energy storage and battery energy storage systems (BESS) to offer cost-competitive renewable energy and load capacity. It was found that a CSP-PV plant with TES and BESS increased the capacity factor, reaching values above 85% [15]. However, a significant reduction in battery bank cost (on the order of 60–90%) was required to achieve significant savings. In current scenarios, a CSP-PV plant with TES presents better economics and reliability compared with a system with BESS [16]. However, in promising future scenarios focused on cost reduction (around USD 60/kWh), the advantages of batteries become evident, and a PV plant with batteries shows a higher competitiveness than a plant with TES. This was also discussed in [17]. Different scenarios were explored in [18] to identify the dominant technology in a hybrid solar power plant providing sustainable and programmable energy by 2050. It was concluded that CSP with TES is currently the most affordable technology. Still, a shift towards PV with BESS is expected, mainly due to both systems' significant reduction in costs.

This paper introduces an innovative integration scheme that combines PV and CSP (based on PTC solar field technology) systems. In this proposed scheme, both solar resources operate synchronously, providing low-temperature heat (PTC) and power (PV) to a novel high-temperature heat pump. This heat pump efficiently transfers high-temperature heat to the power cycle (heat engine), simulating the output that would have been generated by a central receiver system with heliostats, but now with a parabolic trough collector

field. To ensure the system's dispatchability, a two-tank molten-salt TES is strategically positioned between the thermal output of the heat pump and the power cycle. This integration through a heat pump instead of electrical resistors allows the recovery of all the power supplied by the PV, thereby enhancing the thermal supply from the PTC. This fact constitutes a key aspect of the proposal, enabling two main benefits. On the one hand, it enhances both the thermal level and the overall quantity of thermal energy derived from the PTC field. This avoids the need for a central receiver system with a heliostat field to achieve high temperatures and eliminates the necessity for field oversizing. Instead, these functions are fulfilled by the PV field. On the other hand, the coefficient of performance (COP) of the heat pump compensates for the efficiency-related penalty imposed by the power cycle. Consequently, the proposal enables the retrieval of all power supplied by the PV field, a feat not achieved in conventional hybridization, where electrical resistors are used to store PV surplus. Lastly, it is worth noting that the proposed hybridization ensures that all power injected into the grid comes from a synchronous generator, contributing its rotational inertia. This characteristic offers stability advantages compared with direct injection from the PV field.

Figure 2 illustrates the concept of the proposed hybrid plant. The PV power drives the mechanical consumption of the heat pump, while the thermal solar field provides the low-temperature heat input. The heat pump efficiently integrates and converts both energy sources into a high-temperature heat output, which can be either stored or directly converted into electricity by the heat engine. The implementation employs a parabolic trough collector (PTC) solar field to leverage well-established technologies. A two-tank thermal energy storage system, filled with molten solar salt (with a weight composition of 60% $NaNO_3$ and 40% $KNO_3$), operates at temperatures of 589 °C/405 °C. This temperature range aligns with the current state-of-the-art in CSP technology, particularly with central receiver systems [19]. Consequently, this system is anticipated to avoid corrosion and material-related issues, different from those currently reported in CSP plants with storage. The role of the heat pump is to effectively convert the PTC solar field into a virtual heliostat/central tower receiver system, thereby enhancing the overall efficiency of the heat engine.

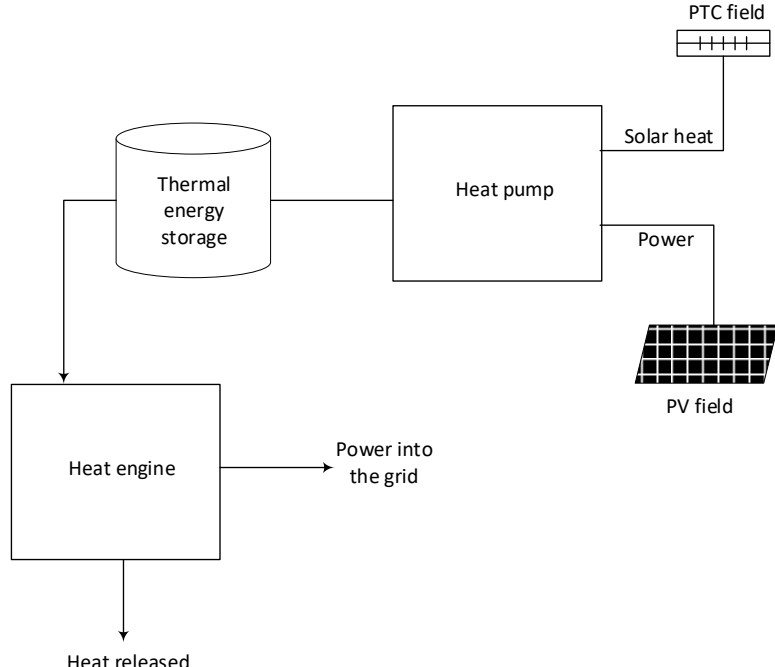

**Figure 2.** Conceptual scheme of the proposed novel hybrid CSP-PV plant.



The core component of this integrated system is a heat pump based on a reverse Brayton cycle, using supercritical $CO_2$ as the working fluid. The fluid's nearly ideal gas behaviour within the working zone, along with the implementation of a turbocompressor instead of a volumetric compressor, enables the attainment of the high temperatures necessary for the molten salt circuits. For the heat engine, a recompression Brayton supercritical $CO_2$ cycle is proposed, leading to high levels of efficiency. Both cycles incorporate heat exchangers connected to the high-temperature reservoir, positioned within the low-pressure stream. This strategic arrangement allows for the use of heat exchangers with enough size to avoid molten-salt clogging issues. Isentropic efficiencies of 92% for turbines and 88% for compressors are established, with temperature approaches ranging between 5 to 10 K and pressure drops of 2% across each heat exchanger stream. The upper pressure limit for both thermal machines is set 300 bar, whereas the lower pressure limit is maintained at 85 bar. These parameters achieve a COP of 2.32 in the heat pump and an efficiency of 44.4% in the heat engine.

## 2. Methodology

### 2.1. Concept

In countries with a high share of intermittent renewable energies (wind and PV), curtailments often occur during hours of solar radiation. Therefore, it is interesting to analyse the scenario where all PV production is stored, considering both conventional and novel hybrid configurations. Equation (1) represents the energy balance in the storage system. It can be applied to both configurations, where $Q$ stands for thermal energy, $W$ represents power, and subscripts *PV*, *SF*, and *TES* denote photovoltaic, solar field, and thermal energy stored, respectively. For the conventional configuration, Equation (2) describes the exergy balance in the storage system, while Equation (3) corresponds to the novel system including the heat pump in the control volume. In these equations, $T$ stands for the average entropic temperature [20] of thermal sources or reservoirs, and $I$ represents exergy destroyed or irreversibility. The subscript 0 refers to the ambient state (dead state), *conv* means conventional, and *nov* means novel.

Comparing Equations (3) and (2) reveals that the novel plant supplies more exergy to the heat engine. This is due to the fact that in the conventional case, the stored energy matches the PTC solar field temperature (380 °C/290 °C), whereas in the novel case, it corresponds to a central tower receiver temperature level (589 °C/405 °C) [19], thanks to the heat pump effect. In other words, the stored energy temperature in the conventional case is the same as the solar field temperature. By combining Equations (2) and (3), Equation (4) is obtained, where *COP* represents the coefficient of performance of the heat pump, and $COP_{max}$ denotes its maximum value, assuming that all processes are totally reversible (Equation (5)). Equation (4) reveals the lower exergy destruction in the heat pump compared to the Joule effect, which is always irreversible.

$$W_{PV} + Q_{SF} = Q_{TES} \tag{1}$$

$$W_{PV} + Q_{SF} \cdot \left(1 - \frac{T_0}{T_{SF}}\right) = Q_{TES} \cdot \left(1 - \frac{T_0}{T_{SF}}\right) + I_{conv} \tag{2}$$

$$W_{PV} + Q_{SF} \cdot \left(1 - \frac{T_0}{T_{SF}}\right) = Q_{TES} \cdot \left(1 - \frac{T_0}{T_{TES-nov}}\right) + I_{nov} \tag{3}$$

$$I_{conv} - I_{nov} = \left(\frac{T_0}{T_{SF}}\right) \cdot \left(\frac{Q_{TES}}{COP_{max}}\right) = \left(\frac{T_0}{T_{SF}}\right) \cdot \left(\frac{COP}{COP_{max}}\right) \cdot W_{PV} \tag{4}$$

$$COP_{max} = \frac{T_{TES-nov}}{T_{TES-nov} - T_{SF}} \tag{5}$$

Figure 3 depicts the proposed layout, providing a detailed overview of the heat pump and heat engine components. The heat pump employs a reverse recuperated Brayton cycle,

while the heat engine utilizes a recompression Brayton power cycle, both with supercritical $CO_2$. In both cases, the heat exchangers that employ molten salts (MSHP in the heat pump and MSHE in the heat engine) have been relocated to the low-pressure side of the cycle. This modification is required because both Brayton cycles operate with a maximum pressure of 300 bar. To handle such a high pressure, printed circuit heat exchangers (PCHE) are typically employed in S-CO2 applications [21]. Based on diffusion bonding, PCHEs can withstand high pressures due to their manufacturing process. However, the channels in these heat exchangers are too narrow, leading to clogging issues when using molten salts. The allocation of molten salt heat exchangers in the proposed layouts enables using a heat exchanger with wide channels for the salt, avoiding clogging issues [22]. One possible option for that kind of heat exchanger might be a hybrid printed circuit heat exchanger, where the layers of one of the streams are replaced with a plate-and-fin structure, becoming a combination of a PCHE and a plate-fin heat exchanger (PFHE). A similar heat exchanger configuration has been proposed for sodium fast reactor (SFR) Generation IV nuclear reactors, where liquid metal cools the reactor by circulating through the plate-fin structure while $CO_2$ flows through the semi-circular channels [23]. These heat exchangers have been available from Heatric since 2006, and are known as hybrid heat exchangers ($H^2X$). They are manufactured using chemically etched sheets and corrugated fins [24]. In the current application, $CO_2$ would flow through the semi-circular channels, while molten salts would flow through the corrugated fins. The corrugated fins are designed to withstand pressures of 200–220 bar [24], making them suitable for their use on the low-pressure side (85 bar [22]).

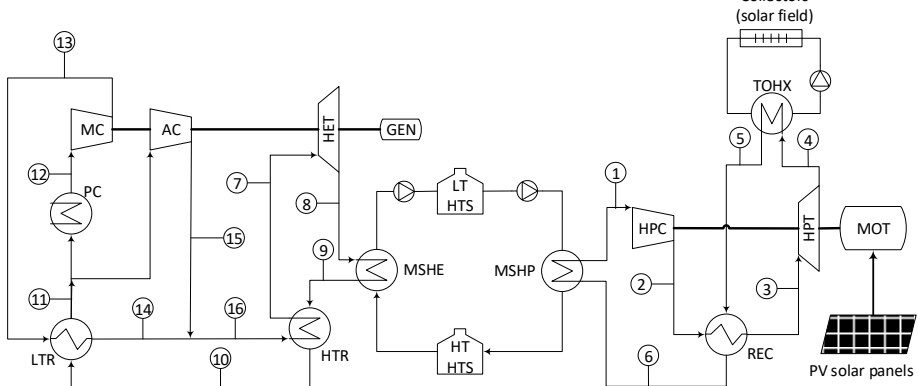

**Figure 3.** Proposed layout.

### 2.2. Heat Pump

The heat pump receives a low-temperature heat input ranging from 300 °C to 390 °C from the thermal oil of the solar field. It then releases high-temperature heat into the molten salt loop, with temperatures ranging from 405 °C to 589 °C. A reverse recuperated Brayton cycle has been adopted to achieve a sensible heat transfer profile in the working fluid of the heat pump. This cycle operates far from the critical point, leading to a nearly ideal gas behaviour.

The cycle can be described as follows (Figure 3): the low-pressure and -temperature stream (4) exiting the turbine (HPT) absorbs heat from the low-temperature reservoir in the thermal oil/$CO_2$ heat exchanger (TOHX). A second stage of heating (5-6) takes place in the recuperator (REC), utilizing the high temperature of the stream leaving the compressor (2-3). Subsequently, the low-pressure stream, now at a high temperature after these two heating stages, provides thermal energy to the hot sink (6-1) via the $CO_2$/molten salt heat exchanger (MSHP). After that, the low-pressure stream enters the compressor (HPC), which increases its pressure and temperature (1-2). After the compressor, the high-pressure stream releases heat in the recuperator (2-3), returning to the turbine inlet conditions (3).

The isentropic efficiencies of the turbine and compressor have been set to 92% and 88%, respectively [25]. The recuperator's temperature approach is assumed to be 5 K, and 10 K in the low-temperature molten salt/$CO_2$ heat exchanger. The compressor's outlet pressure is 300 bar, with an inlet pressure of 85 bar and a temperature of 415 °C. Pressure drops of 2% have been considered for each stream in the heat exchangers [21]. Table 1 outlines the characteristics of the state points for the heat pump. To visually depict this information, Figure 4 illustrates the p-h diagram. Notably, the behaviour of the fluid might be approximated to that of an ideal gas operating in a reverse Brayton cycle.

**Table 1.** Properties at state points (Figure 3) of the heat pump.

| Point | Pressure [bar] | Temperature [°C] | Enthalpy [kJ/kg] |
|:-----:|:--------------:|:----------------:|:----------------:|
| 1 | 85.00 | 415.0 | 377.3 |
| 2 | 300.0 | 604.1 | 590.8 |
| 3 | 294.0 | 388.5 | 318.3 |
| 4 | 90.31 | 261.4 | 199.6 |
| 5 | 88.5 | 370.0 | 324.5 |
| 6 | 86.73 | 599.1 | 596.9 |

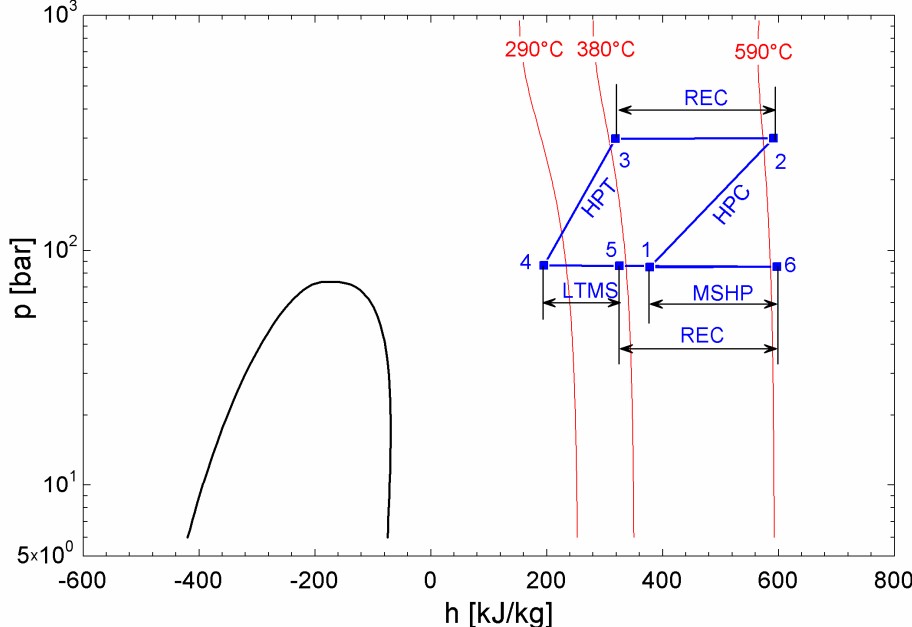

**Figure 4.** Pressure-enthalpy diagram of the heat pump.

### 2.3. Heat Engine

A Brayton supercritical $CO_2$ power cycle has been selected for the heat engine [21], recommended by the National Renewable Energy Laboratory of the US for future concentrated solar power plants [26]. This power cycle is chosen mainly for its compactness and efficient operation at moderate temperatures. The high efficiencies achieved by S-CO2 power cycles [21] are due to the proximity of the compressor inlet to the critical point of $CO_2$ (31.1 °C and 73.8 bar), which reduces the work required for compression. However, that proximity can cause density variations, which can be managed by setting the compressor inlet pressure at 85 bar to strike a balance between stable operation and reduced compression energy consumption [22]. Another challenge that may arise in the compressor is the occurrence of phase-change phenomena resulting from local flow accelerations. Operating in the two-phase region can negatively impact compressor performance, causing erosion and degradation of certain components. To carefully design the main compressor while considering these challenges is crucial to prevent such issues and ensure the reliable functioning of the power cycle.

Furthermore, working close to the critical point also poses a problem in a simple recuperated Brayton cycle. The specific heat of $CO_2$ in the low-pressure stream of the recuperator is lower than that in the high-pressure stream, resulting in an imbalanced temperature profile that prevents achieving maximum performance in the heat recovery process. This problem is commonly addressed by dividing the recuperator into two units, leading to the implementation of a re-compression cycle [21].

Based on Figure 3, which illustrates the heat engine employed in this study, the $CO_2$ exiting the turbine (HET) absorbs thermal energy from the molten salts in the molten salt/$CO_2$ heat exchanger (MSHE) (8-9) and transfers it to the cold stream (16-7) in the high-temperature recuperator (HTR), reaching the turbine's inlet condition. After leaving the HTR, the hot stream once again transfers thermal energy (10-11) in the low-temperature recuperator (LTR) to the cold stream (13-14). Notably, the mass flow rate of the cold stream in the LTR (high-pressure stream) is lower than that of the hot stream (low-pressure stream). This factor is crucial in recompression cycles to achieve balance in the LTR since the high-pressure stream has a higher specific heat than the low-pressure stream. The division of mass flow rate occurs at the inlet (11) of the precooler (PC). Then, the main flow undergoes cooling (11-12) and compression (12-13) in the main compressor (MC), while the remaining flow (14-15) is directly sent to the auxiliary compressor (AC). As this stream is compressed without previous cooling, it reaches an appropriate temperature to be mixed with the cold stream leaving the LTR (14-15-16), forming the cold stream that enters the HTR.

The turbine's isentropic efficiency is assumed to be 92%, and an efficiency of 88% is considered for both compressors [25]. A temperature approach of 5 K is employed in the HTR and 5.5 K in the LTR. The approach temperature in the MSHE has been set to 10 K. The outlet pressure of the main compressor is set to 300 bar, with an inlet temperature of 35 °C and an inlet pressure of 85 bar. Pressure drops of 2% are considered for each stream of the heat exchangers [21]. Table 2 outlines the characteristics of the state points within the heat engine. To provide a visual representation of these points, Figure 5 illustrates the p-h diagram. Notably, the compression processes within the heat engine exhibit steeper slopes, especially in the main compressor, due to their operation in the proximity to the critical point. It is worth emphasizing that the enthalpy change in the LTR reveals a greater length for the high-pressure stream in comparison with the lower-pressure stream. This discrepancy is a consequence of the lower mass flow ratio in the former, which compensates for its higher specific heat.

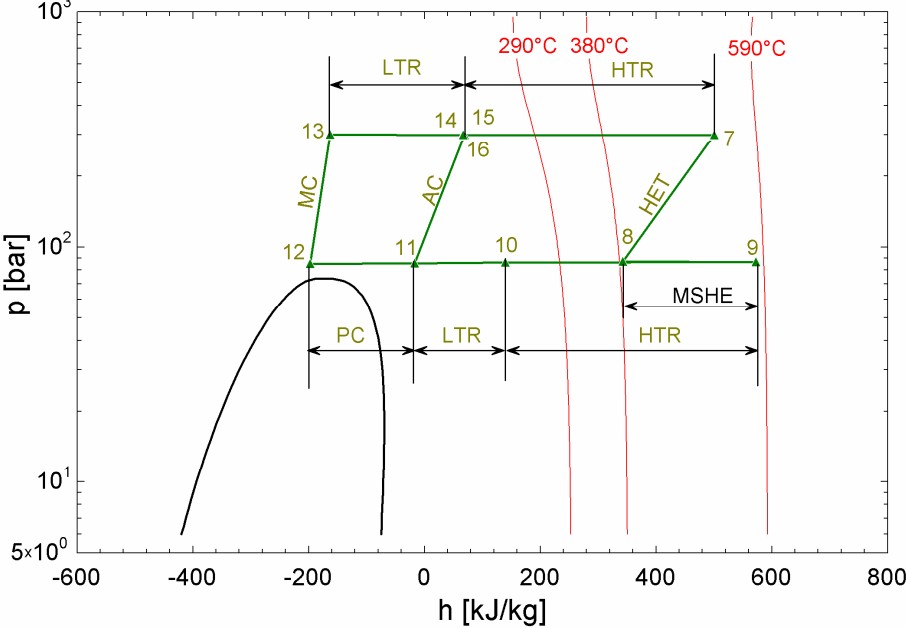

**Figure 5.** Pressure-enthalpy diagram of the heat engine.

**Table 2.** Properties at state points (Figure 3) of the heat engine.

| Point | Pressure [bar] | Temperature [°C] | Enthalpy [kJ/kg] |
|---|---|---|---|
| 7 | 288.1 | 532.8 | 501.0 |
| 8 | 92.15 | 395.8 | 353.8 |
| 9 | 90.31 | 579.1 | 572.3 |
| 10 | 88.50 | 203.6 | 133.9 |
| 11 | 86.73 | 82.02 | −18.61 |
| 12 | 85.00 | 35.00 | −197.9 |
| 13 | 300.0 | 76.52 | −163.3 |
| 14 | 294.0 | 198.1 | 61.88 |
| 15 | 294.0 | 199.6 | 64.25 |
| 16 | 294.0 | 198.6 | 62.65 |

*2.4. Solar Field*

The solar field consists of parabolic trough collectors that use Therminol VP1 [27] as the heat transfer fluid. This choice of fluid is well established and widely adopted for the specific temperature range under consideration [19].

To determine the solar field's appropriate sizing and nominal performance, the design point has been selected as the 21st of June at solar noon in Seville (Spain). At this point, the direct normal irradiation (DNI) is set to 900 W/m$^2$. The specific type of parabolic trough collector employed is the Eurotrough, detailed in Table 3 [28]. The thermal performance calculation considers the heat losses correlation provided by the manufacturer [29].

**Table 3.** Geometric and optical parameters of the solar field [28].

| | |
|---|---|
| Number of loops in the solar field | 78 |
| Number of collectors per loop | 4 |
| Number of modules per collector | 10 |
| Length of every module [m] | 12.27 |
| Absorber tube outer diameter [m] | 0.07 |
| Absorber tube inner diameter [m] | 0.065 |
| Glass envelope outer diameter [m] | 0.115 |
| Glass envelope inner diameter [m] | 0.109 |
| Intercept factor | 0.92 |
| Mirror reflectivity | 0.92 |
| Glass transmissivity | 0.945 |
| Solar absorptivity | 0.94 |
| Peak optical efficiency | 0.75 |

Table 3 shows that there are a total number of 78 loops in the system. Each loop comprises 4 collectors, and each collector consists of 10 modules. The overall length of each loop is 490.8 m. Table 4 provides a summary of the key parameters that characterise the nominal performance of each loop.

**Table 4.** Nominal performance of the parabolic trough loop.

| | |
|---|---|
| Mass flow per loop [kg/s] | 7.6 |
| Inlet/outlet HTF temperature [°C] | 300/390 |
| Inlet pressure [bar] | 20 |
| Heat gain per loop [MWth] | 1.6732 |
| Heat loss per loop [kWth] | 158.56 |
| Pressure drop per loop [bar] | 4.1438 |
| Optical efficiency [%] | 71.99 |
| Thermal efficiency [%] | 91.34 |

### 2.5. Sizing of Heat Exchangers

In the heat pump and heat engine, two types of heat exchangers have been used: hybrid ($H^2X$) and printed circuit heat exchangers (PCHE). As explained earlier, $H^2X$ prevents potential clogging problems, while PCHEs are commonly employed in S-CO2 applications to withstand high pressure differences.

An iterative design approach has been used to perform the sizing of the heat exchangers, based on the sub-heat exchanger methodology. Sizing calculations are greatly influenced by the significant changes in $CO_2$ properties, particularly when the operating conditions are close to the critical point. This method assumes a continuous variation of properties throughout the sub-heat exchangers that comprise the entire unit [30].

Using the information supplied by the leading manufacturer of the PCHEs [31], the maximum plate size is 600 × 1500 mm. Modules are built with the plates, assuming a core block with a maximum height of 600 mm. This configuration results in a total of 96,000 channels (48,000 per stream). This kind of design comes from the manufacturer [32], and the authors have previously employed it in a CSP project [22]. The channels where the fluid flows are semicircles with a 2 mm diameter and a 2.5 mm pitch. The thickness of the plates is 1.6 mm. Inconel alloy 617 has been recommended for both the receiver (REC) and high-temperature recuperator (HTR) due to the elevated temperatures [33]. Austenitic stainless steel 316L has been selected for the remaining PCHEs.

The overall heat transfer coefficient for each sub-heat exchanger has been evaluated following the methodology described by Dostal [21]. The length of the heat exchanger is determined using the heat transfer across the surface area; consequently, the pressure drop along the entire channel is assessed for every stream. The total number of channels is iterated during the design process until the desired temperatures at each port are achieved. If the calculated maximum pressure drop exceeds or falls below the desired value, the number of tubes increases or decreases accordingly.

### 2.6. Economic Model

The estimated investment cost (fixed capital investment, FCI, based on [34]) has been determined. The total cost can be split into direct and indirect costs, with indirect costs considered to be 25% of the direct costs. Indirect costs encompass engineering, supervision, construction, and contingencies. Direct costs include on-site costs (piping instrumentation and controls, equipment) and off-site expenses (civil works, land and service facilities). For this study, the off-site costs have been incorporated into the solar field and the storage systems, which are directly accounted for in the direct costs. The remaining portion of the facility's cost is determined based on the purchased equipment cost (PEC) by multiplying it by a factor of 2.18, taken from a study from Sandia National Laboratory [35] for a recompression Brayton supercritical $CO_2$ cycle of 10 MWe [35]. This study has been also used to estimate the cost of the heat engine and heat pump cycles.

The PCHEs' costs are obtained by scaling them according to the number of modules. An escalation factor of 0.4 [34] is applied, with a base PEC of USD 5 M for the HTR and REC, and a number of modules of 4.46. For the remaining PCHEs, the base PEC is assumed as USD 3 M and a number of modules of 3.1. These cost figures are based on data from 2013 [35]. The cost difference in the PCHEs is due to the different operating conditions in the HTR and REC and the rest of the PCHEs: the HTR and REC operate at high temperatures, necessitating a high-strength alloy such as Inconel 617. In contrast, the other heat exchangers can be manufactured using a standard SS 316 material [36]. The sizing of the $H^2X$ heat exchangers has yet to be investigated. Instead, the specific cost (USD/kW) obtained for PCHEs within a similar temperature range has been considered to estimate the investment required.

Equations (6)–(9) represent the escalation process required to estimate the investment in turbomachinery, in the generator of the heat engine and the motor in the heat pump, respectively, taken from [37]. In these equations, *W* stands for the electricity power (produced by the heat engine and consumed by the heat pump) in MW, *p* for the highest pressure

in bar, and $T$ for the highest temperature in Celsius degrees. Finally, the *PEC* of the set is given in USD million.

$$f_W = \left(\frac{W}{10}\right)^{0.68} \tag{6}$$

$$f_p = \left(\frac{p}{200}\right)^{-0.6} \tag{7}$$

$$f_T = \frac{3.35 + \left(\frac{T}{1000}\right)^{7.8}}{3.35 + \left(\frac{650}{1000}\right)^{7.8}} \tag{8}$$

$$PEC = f_W \cdot f_p \cdot f_T \cdot 6 \tag{9}$$

The thermal energy storage cost has been estimated from the NREL Gen3 roadmap for CSP [26], considering a two-tank solar salt system from Abengoa as a reference. The costs provided include both on-site and off-site expenses, representing direct costs. The volume of the cold tank serves as the baseline (15,700 m³), and an escalation factor of 0.8 is used. The base cost breakdown is: hot tank, USD 10.016 M; cold tank, USD 4.361 M; tank insulation, USD 3.724 M; foundations, USD 3.050 M; structural steel, USD 0.666 M; electrical installations, USD 0.481 M; site works, USD 0.339 M; and instrumentation, USD 0.212 M. The cost of the salt inventory varies linearly and is calculated at a specific cost of USD 1100/tonne. For the solar field investment cost estimation, a specific cost of USD 152 per solar aperture area has been considered, with reference to 2020 [38].

A time scaling technique is applied using the Chemical Engineering Plant Cost Index (CEPCI) to account for data from different dates. The CEPCI value for 2013 (applicable to molten salt loops, PCHEs, and turbomachines) is 567.3, while the value for 2020 (the year the results are referenced) is 596.2.

A preliminary calculation of the LCOE is performed, considering just the known investment, previously determined. Equation (10) allows the calculation of the LCOE, where *CRF* refers to the capital recovery factor, as defined in Equation (11), $\dot{W}_{HE}$ stands for the net power of the heat engine, and $H_{HE}$ represents the equivalent operating hours of the heat engine per day at constant load. In Equation (11), the variable *wacc* denotes the weighted average capital cost, assumed to be 7.5%, and $N$ represents the project's lifespan, in this case set at 25 years. Concerning the investment in the PV field, an LCOE of EUR 25/MWh [3] is used, with a production of 1800 equivalent hours per year.

$$LCOE = \frac{FCI \cdot CRF}{\dot{W}_{HE} \cdot H_{HE} \cdot 365} \tag{10}$$

$$CRF = \frac{wacc \cdot (1 + wacc)^N}{(1 + wacc)^N - 1} \tag{11}$$

## 3. Results

Equations (12) and (13) provide the expression for the efficiency ($\eta_{HE}$) of the heat engine and the *COP* of the heat pump, respectively. The subscripts in these equations correspond to the same acronyms used for the cycle components. The ratio between the mass flow rate of the main compressor and the turbine of the heat engine is determined to be 0.6772, derived from the LTR balance (same terminal temperature difference at both extremes). It is important to note that the product of the heat engine efficiency and the coefficient of performance (Equation (14)) represents the ratio of energy production to the energy input from the PV field. This value is slightly greater than 1. This result allows considering the proposed plant as a PV field with storage with certain gain [39]. In the design of the proposed plant, a pressure drop of 2% has been considered in each heat exchanger stream, in order to limit its size. If a lower pressure drop had been chosen, the

ratio defined by Equation (14) would have been greater [39]. Thus, the recovery of the PV power supplied to the heat pump is possible with the chosen technology.

$$\eta_{HE} = \frac{W_{GEN}}{Q_{MSHE}} = \frac{(h_7 - h_8) - \left(\frac{\dot{m}_{MC}}{\dot{m}_{HET}}\right) \cdot (h_{13} - h_{12}) - \left(\frac{\dot{m}_{AC}}{\dot{m}_{HET}}\right) \cdot (h_{15} - h_{11})}{h_9 - h_8} = 44.4\% \quad (12)$$

$$COP = \frac{Q_{MSHP}}{W_{MOT}} = \frac{h_6 - h_1}{(h_2 - h_1) - (h_3 - h_4)} = 2.32 \quad (13)$$

$$\eta_{HE} \cdot COP = \frac{W_{GEN}}{W_{MOT}} = 1.03 \quad (14)$$

The energy balance of the plant is presented in Tables 5 and 6. The sizing of the solar field, with a capacity of 128 MWth, is based on the dimensions of existing CSP plants in Spain that utilize parabolic trough collectors without storage [19]. As a reference, these plants produce 50 MWe. To ensure the ability to evacuate a portion of the power during production while storing the remainder for later use, the MSHE heat exchanger has been designed to be half the size of the MSHP. Thus, half of the thermal energy discharged by the heat pump is stored in the molten salt tanks. However, for the sake of flexibility, the storage capacity of the tanks has been doubled, being able to store all the heat discharged by the heat pump. This configuration enables the plant to prevent the risk of curtailments.

**Table 5.** Energy balance at the design point of the heat pump (Figure 2). In the heat exchangers, the hot stream is designed in the first place.

| Component | Heat Duty or Power [MW] | Mass Flow Rate [kg/s] |
|---|---|---|
| Compressor (HPC) | 219 | 1025 |
| Turbine (HPT) | 122 | 1025 |
| Motor (MOT) | 97 | --- |
| $CO_2/CO_2$ (REC) | 279 | 1025/1025 |
| Thermal Oil/$CO_2$ (TOHX) | 128 | 582/1025 |
| $CO_2$/Molten salts (MSHP) | 225 | 1025/798 |

**Table 6.** Energy balance at the design point of the heat engine (Figure 2). In the heat exchangers, the hot stream is designed in the first place.

| Component | Heat Duty or Power [MW] | Mass Flow Rate [kg/s] |
|---|---|---|
| Main Compressor (MC) | 12 | 349 |
| Auxiliary Compressor (AC) | 14 | 167 |
| Turbine (HET) | 76 | 516 |
| Generator (GEN) | 50 | --- |
| $CO_2/CO_2$ (HTR) | 226 | 516/516 |
| $CO_2/CO_2$ (LTR) | 779 | 516/349 |
| Molten salts/$CO_2$ (MSHE) | 113 | 399/516 |
| $CO_2$/Water (PC) | 63 | 349/2995 |

Figures 6 and 7 compare the performance of conventional hybrid CSP-PV plant with a novel hybrid design under two scenarios: design energy storage (a) and maximum storage (b). To ensure a valid comparison, the sizes of the PV and PTC fields are identical in both cases, determined based on the configuration of the novel design. To simplify the analysis, a solar radiation period of 6 equivalent hours has been assumed.

In Figure 6a, the conventional hybrid CSP-PV plant produces 97 MW of power from the PV field and stores 128 MW of thermal energy from the PTC field during the 6 h of radiation. Subsequently, in the next 6 h, 50 MW of power is produced through the

discharge of molten salts through the heat engine. This results in an overall production of 882 MWhe. In Figure 7a, the novel hybrid CSP-PV plant generates 50 MW of power during both radiation and non-radiation periods, summing up to 600 MWhe. While the electricity production is 282 MWhe lower in the novel design, the entire power injected into the grid comes from a synchronous generator, providing rotational inertia. Conversely, in the conventional plant, 582 MWhe come directly from the PV field.

When curtailments occur, it is necessary to store all generated energy for subsequent injection into the grid. This scenario is analysed in Figure 6b for the conventional hybrid CSP-PV plant and in Figure 7b for the novel design. In Figure 6b, both the thermal output from the PTC field (128 MW) and the power converted from PV through electrical resistors (97 MW) are stored as thermal energy in molten salts during the radiation period. This stored thermal energy is then utilized through the conventional Rankine cycle (50/128 efficiency), resulting in a production of 527.34 MWhe (50 MW over 10.55 h). Regarding Figure 7b, the entire thermal output of the heat pump (225 MW) is stored within the molten salts during the radiation period. Subsequently, this stored thermal energy is converted into power (50 MW) through the S-CO2 power cycle during the discharge period (12 h), resulting in an injection of 600 MWhe into the grid. In this scenario, the electricity production originates from a synchronous generator in both plant configurations. However, the conventional plant experiences a loss of 72.66 MWhe due to the inability of the Rankine power cycle's efficiency penalty to be offset by any heat pump.

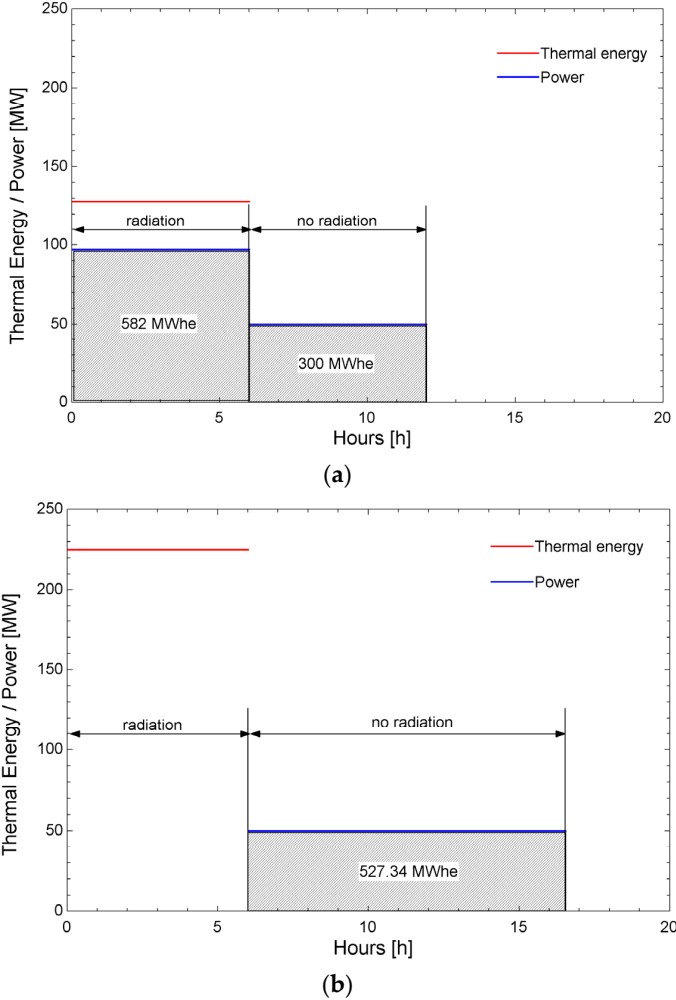

**Figure 6.** Performance of conventional hybrid plant: (**a**) No storage of PV production; (**b**) PV production is fully stored.

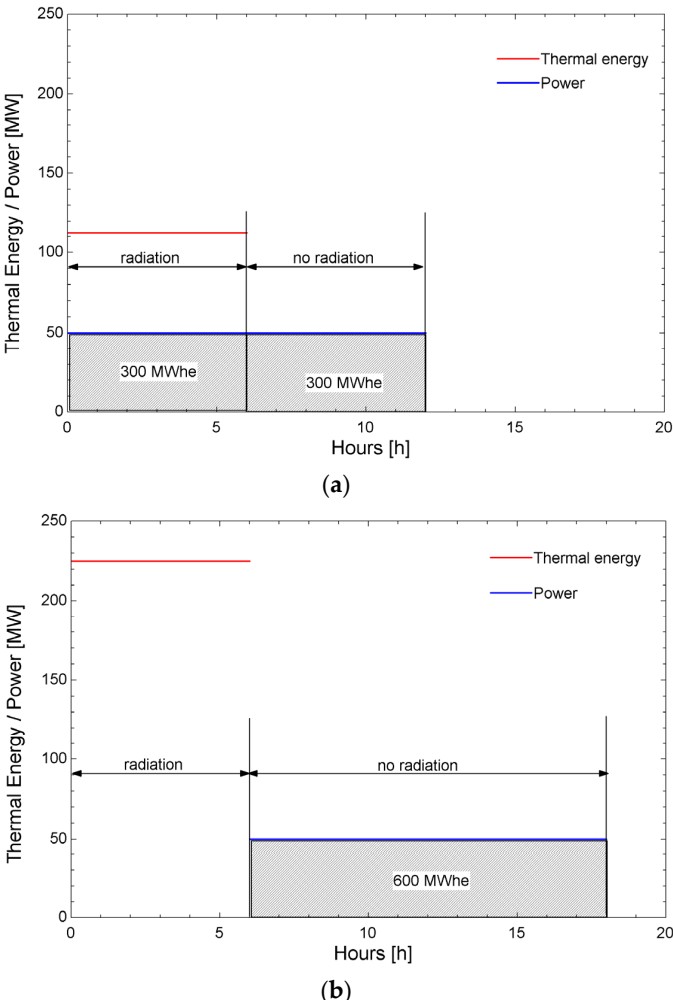

**Figure 7.** Performance of novel hybrid plant: (**a**) half of solar radiation is stored; (**b**) the entire solar radiation is stored.

Tables 7–10 present the cost breakdown of different components of the proposed plant. By aggregating all on-site and direct costs and accounting for 25% of indirect costs, a total fixed capital investment of USD 416.5 M is obtained (equivalent to USD 8329/kWe). In order to facilitate a comparison of this investment with current technologies, Table 11 provides the specific costs of the main subsystems.

**Table 7.** Size and costs of PCHE.

| Component | Heat Duty [MW] | Height [m] | Number of Modules | On-Site Cost [USD M$_{2020}$] |
|---|---|---|---|---|
| REC | 279 | 3.92 | 96 | 39.0 |
| HTR | 226 | 2.67 | 22 | 21.7 |
| LTR | 79 | 4.25 | 32 | 17.5 |
| PC | 63 | 0.43 | 11 | 11.4 |
| TOHX | 128 | 0.40 | 8 | 10.2 |

**Table 8.** Costs of H$^2$X.

| Component | Heat Duty [MW] | Reference PCHE | On-Site Cost [USD M$_{2020}$] |
|---|---|---|---|
| MSHP | 225 | REC | 30.0 |
| MSHE | 113 | HTR | 10.3 |

**Table 9.** Turbomachine and electrical machine costs.

| Cycle | On-Site Cost [USD M$_{2020}$] |
|---|---|
| HP | 50.3 |
| HE | 32.7 |

**Table 10.** Molten salt loops, PTC and PV costs.

| Component | Energy Stored [MWh] | Salt Inventory [ton] | Direct Costs [USD M$_{2020}$] |
|---|---|---|---|
| TES | 1351 | 17,245 | 38.6 |
| PV field | --- | --- | 38.9 |
| PTC Field | --- | --- | 32.5 |

**Table 11.** Specific costs of the main components.

| Component | Base Parameter | Specific Cost (FCI) |
|---|---|---|
| HP [USD$_{2020}$/kWt] | Heat released (225 MW) | 719.44 |
| HE [USD$_{2020}$/kWe] | Power produced (50 MW) | 2340 |
| TES [USD$_{2020}$/kWht] | Thermal energy stored (1351 MWh) | 35.71 |
| PV field [USD$_{2020}$/kWp] | Peak power (97 MW) | 501.3 |
| PTC field [USD$_{2020}$/m$^2$] | Solar aperture area (21.7 hm$^2$) | 190.0 |

Accurately determining the LCOE requires an annual simulation, taking into account the off-design response of the entire system. Such an assessment is beyond the scope of the current work. Instead, a preliminary assessment of the LCOE has been carried out, considering a daily production of 600 MWh (6 equivalent hours throughout the year). This results in an LCOE of USD 171/MWhe.

## 4. Discussion and Conclusions

The integration of a PV plant and a CSP through a high-temperature heat pump has been analysed. This configuration introduces a novel approach by employing a heat pump as the integration mechanism instead of conventional electrical resistors found in existing CSP-PV plants.

For the proposed system, a recompression Brayton supercritical $CO_2$ cycle is used as the heat engine, whereas a reverse recuperative Brayton cycle, also operating with supercritical $CO_2$, is employed for the heat pump. The thermal energy storage system incorporates a two-tank configuration utilizing solar salt as the medium for storing sensible heat. The solar field consists of parabolic trough collectors, which serve as the cold source for the heat pump. It is important to highlight that the PV plant exclusively drives the heat pump, with no direct injection of PV power into the grid.

The heat pump allows for storing high-grade thermal energy in the molten salts using a parabolic trough collectors field, as if it were produced in a heliostat solar field. Moreover, the integration through the heat pump reduces exergy losses in the storage system. According to Equation (3), and considering the average entropic temperatures [20] of the PTC field (616.9 K) and the molten salts (766.3 K), the exergy loss reduction (calculated in Equation (15)) for the storage system amounts to 45.24%.

$$\frac{I_{conv} - I_{nov}}{I_{conv}} = \frac{COP}{COP_{max}} = \frac{2.32}{5.13} = 45.24\% \tag{15}$$

The high-temperature heat pump operates within the nearly ideal gas region as depicted in Figure 4. This characteristic significantly reduces the technological risk associated with the reverse recuperated Brayton cycle. On the other hand, the heat engine, which

utilizes the recompression Brayton supercritical $CO_2$ cycle, has limited commercial experience. In early demonstration plants, it could be replaced by a conventional Rankine cycle, although losing efficiency. The proposed S-CO2 cycle achieves an efficiency of 44.4%, surpassing the less than 40% observed in current CSP plants [19].

The performance of hybrid CSP-PV plants has been analysed under two different scenarios: design conditions (Figures 6a and 7a) and full storage (Figures 6b and 7b). In the former, the conventional scheme injects 47% more electricity into the grid during a typical day compared to the novel scheme. However, the injection during radiation hours lacks rotational inertia in the conventional scheme. In the latter scenario, where full storage is required, the novel scheme injects 13.8% more electricity into the grid than the conventional one, with both injections having rotational inertia. In this case, the use of electrical resistors instead of a heat pump results in higher irreversibilities, which reduces the recovered power.

In terms of costs, the preliminary estimation of the LCOE has been based on an assumption of 6 equivalent hours of operation each day throughout the year, resulting in a total production of 12 h, equivalent to 50% of the capacity factor. The obtained LCOE (USD 171/MWh) is lower than the current CSP technology with a 45% capacity factor [40]. In [22], Generation 3 technology is examined, based on S-CO2 with turbine inlet temperature up to 800 °C of, ternary salts, a 38% capacity factor, and central tower receivers. The resulting LCOE for this technology is USD 238/MWh. This demonstrates the significant cost advantage of the heat pump system, which replaces the expensive heliostat solar field with a combination of a PV plant and a parabolic trough collector field.

Figure 8 provides a detailed breakdown of the costs associated with this novel system. It reveals that more than half of the total cost is attributed to the PCHE and turbomachines. This fact suggests that there is potential for cost reduction, particularly in these components. Notably, a cost reduction is expected as the S-CO2 technology deploys, because these components, especially the heat engine compressors, are currently in the first steps of the learning curve.

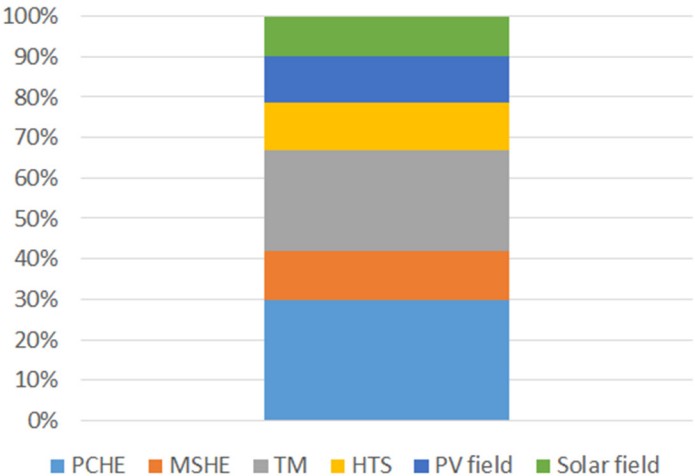

**Figure 8.** Cost breakdown of the novel hybrid CSP-PV plant.

If we consider the same energy production of 600 MWh per day, but with a 100 MW capacity PV plant and 6 h of battery storage, the LCOE would be lower than USD 140/MWh (assuming a battery cost of USD 250/kWh [41] and total battery replacement over the lifespan of the project). However, it is important to acknowledge that this battery system presents two significant drawbacks. Firstly, it requires a substantial amount of critical raw materials. Secondly, it results in the loss of rotational inertia, necessitating additional controls to manage grid frequency. In contrast, the proposed system injects electricity into the grid through a synchronous generator in the heat engine, without relying on critical raw materials for its storage components.

Although the proposed system has not been able to beat the cost of a PV with battery storage, it exhibits several operational and construction advantages. In conclusion, this proposed system effectively reduces the LCOE compared with both current CSP power plants. Additionally, it enhances the performance of current hybrid CSP-PV power plants, especially in terms of long-duration energy storage capabilities.

The preliminary LCOE estimation conducted in this study should be further refined in future works through annual simulations, including off-design responses of the entire system. While the current state of the art has been considered wherever possible (such as PTC and molten salts), the potential for evaluating advanced subsystems remains open for future exploration. Similarly, different S-CO2 layouts could be analysed, alongside current steam Rankine power cycles used in CSP, thereby establishing a comprehensive roadmap towards achieving the final hybrid configuration.

**Author Contributions:** Conceptualization, J.I.L., A.M.-C. and E.A.; Formal analysis, E.A. and M.J.M.; Funding acquisition, E.A.; Investigation, J.I.L. and A.C.; Methodology, E.A., M.J.M. and A.C.; Project administration, J.I.L. and E.A.; Resources, E.A.; Software, A.C.; Supervision, J.I.L.; Validation, J.I.L., A.C. and J.R.P.-D.; Visualization, J.R.P.-D.; Writing—original draft, J.I.L. and A.M.-C.; Writing—review & editing, E.A., A.C. and J.R.P.-D. All authors have read and agreed to the published version of the manuscript.

**Funding:** This research was funded by Rafael Mariño Chair on New Energy Technologies of Comillas Pontifical University.

**Institutional Review Board Statement:** Not applicable.

**Informed Consent Statement:** Not applicable.

**Data Availability Statement:** Data sharing not applicable.

**Conflicts of Interest:** The authors declare no conflict of interest. The funders had no role in the design of the study; in the writing of the manuscript; or in the decision to publish the results.

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
