# Peer review of "A Novel Hybrid CSP-PV Power Plant Based on Brayton Supercritical CO2 Thermal Machines"

_applsci, doi:10.3390/app13179532_

Round 1

Reviewer 1 Report

This manuscript presents a novel hybrid CSP-PV power plant. The current manuscript is generally complete and reasonable, but there are still some comments to improve the quality of this paper :

1、The purpose and significance of the study needs to be described more clearly.What is the scientific value and new contribution of current research?

2、The introduction needs to be more brief and logical, focusing on the part of the research that is highly relevant.

3、The work of this study includes the experiments taken and the parameters of the study need to be outlined in the last paragraph of the introduction..

4、The experimental setup of the system as a whole needs to be presented more clearly.

5、In addition to the comparison of new hybrid power stations and conventional power stations, the advantages of molten salt relative to resistance charging can be highlighted.

6、The analysis of the results should be more detailed, and the advantages of the new system are not convincing enough in the analysis of the conclusions.

7、The limitations and outlook of this study should be mentioned in the conclusion.

8、Although the research presents the performance advantage with the new hybrid system, additional factors, such as Installation and technology, also need to be considered.

9、As a scientific work, current research may not be enough to meet the journal's requirement.

10、For the evaluation of system performance, it would be better if there were more comprehensive indicators.

11、The structure of the literature can also be optimized, such as making the results and conclusions clearer, and distinguishing the experimental settings from the evaluation indicators.

No

Reviewer 2 Report

I think this paper has low novelty and can not be published in the journal of applied sciences. Therefore, I reject it.

Some revisions are required.

Reviewer 3 Report

The manuscript is interesting and well prepared in general but I have two major points of critique:

1) The authors are using a static method to calculate the LCOE with the assumption of a certain number of annual or daily full load hours. This is not appropriate for solar plants or other renewable plants with intermittent inputs. SolarPACES recommends the simualtion of a typical year with hourly time resolution instead.

2) A table with specific costs for all subsystems should be included to ease comparison to other studies. E.g.
PV field: $/kW_peak
PTC field : $/m² of aperture area 
etc.

Further remarks:

- Line 133: "014]" should rather be "[14]"

- Equation (3): "I_con" should rather be "I_conv"

- Equation (14): From this number the system looks very promising since it could be interpreted as round trip efficiency. But this number dependes on the assumptions made in 2.2 and if these inputs are somewhat lower, the RTE falls below 1.0 which implies that direct feed in of PV electricity would be more economic. This should be discussed.

- Fig. 4: This figure is very confusing for me. A larger storage should give less energy output per day for conventinal CSP-PV-plants? I would expect the opposite. This should be explained.

- Line 555 ff: Comparison of own LCOE figures with others from literature are critical since they are only comparable if the same assumptions and calculation methods were applied.

Round 2

Reviewer 2 Report

The authors tried to improve the manuscript and several figures and notes are added to the manuscript. Except for minor language revisions, the manuscript can be published.

Minor revision is needed.